# Towards Interpretable Chit-chat: Open Domain Dialogue Generation with Dialogue Acts

## Abstract

Conventional methods model open domain dialogue generation as a black box through end-to-end learning from large scale conversation data. In this work, we make the first step to open the black box by introducing dialogue acts into open domain dialogue generation. The dialogue acts are generally designed and reveal how people engage in social chat. Inspired by analysis on real data, we propose jointly modeling dialogue act selection and response generation, and perform learning with human-human conversations tagged with a dialogue act classifier and a reinforcement approach to further optimizing the model for long-term conversation. With the dialogue acts, we not only achieve significant improvement over state-of-the-art methods on response quality for given contexts and long-term conversation in both machine-machine simulation and human-machine conversation, but also are capable of explaining why such achievements can be made.

## 1 Introduction

Conversational agents are becoming ubiquitous recently. Through human-machine conversation, such agents either help users complete specific tasks (Young et al., 2013) or engage them in social chat (Vinyals & Le, 2015). Depending on application scenarios, various conversational agents have been designed including chatbots, personal assistants, and automated customer service, etc.

Traditional research on conversational agents focuses on task-oriented dialogue systems (Young et al., 2013) where task specific dialogue acts are handcrafted in a form of slot-value pairs. On the one hand, through slot-filling, the dialogue acts make conversations in such systems interpretable and controllable; on the other hand, they also hinder scaling such systems to new domains. To escape from the limitation, recent interest of research moves to end-to-end dialogue learning without any assumptions on dialogue acts. Most of the effort is paid to non-task-oriented chit-chat (Vinyals & Le, 2015), and there are also a few studies on task-oriented dialogues (Bordes & Weston, 2017; Eric & Manning, 2017). Without dialogue acts, these work directly constructs a response by learning from large scale data with neural networks, and thus is easy to scale to new domains. On the other hand, due to the absence of dialogue acts, it is hard to interpret the emergence of a response to a dialogue context and predict where the conversation will flow to.

In this work, we aim to achieve interpretability and controllability in non-task-oriented dialogues. To this end, we introduce dialogue acts into open domain dialogue generation. Open domain dialogue generation has been widely applied to chatbots which aim at engaging users by keeping conversation going. Existing work concentrates on generating relevant and diverse responses for a static context. However, it is not clear if relevance and diversity are sufficient to engagement in dynamic interactions. Therefore, we investigate the following problems: (1) if we can properly design dialogue acts that can enable us to understand engagement in human-human open domain conversation; (2) how to learn a dialogue generation model with the dialogue acts; and (3) how the model performs in practice and if the performance can be explained by the dialogue acts.

To examine how people engage in social chat, we establish a general dialogue act taxonomy for open domain conversation by extending the existing work with high-level dialogue acts regarding to conversational context. The taxonomy, when applied to real data, gives rise to an interesting finding that in addition to replying with relevance and diversity , people are used to driving their social chat by constantly switching to new contexts and properly asking questions. Such behaviors are less explored before, and thus are difficult for the existing end-to-end learning methods to imitate. To

mimic human behaviors, we propose jointly modeling dialogue act selection and response generation in open domain dialogue generation. The dialogue model is specified with neural networks. We propose learning from human-human interactions by fitting the model to large scale real world dialogues tagged with a dialogue act classifier and further optimizing the policy of act selection for long-term conversation through a reinforcement learning approach. Our model enjoys several advantages over the existing models: (1) the dialogue acts provide interpretation to response generation from a discourse perspective; (2) the dialogue acts enhance diversity of responses by expanding the search space from language to act $\times$ language; (3) the dialogue acts manage the flow of human-machine conversations and thus enhance human engagement; and (4) the dialogue act selection is compatible with post-engineering work (e.g., combination with rules), and thus allows engineers to flexibly control their systems through picking responses from their desired dialogue acts. Evaluation results on large scale test data indicate that our model can significantly outperform state-of-the-art methods in terms of quality of generated responses regarding to given contexts and lead to long-term conversation in both machine-machine simulation and human-machine conversation in a way similar to how human behave in their interactions.

Our contributions in this work include: (1) design of dialogue acts that represent human behavior regarding to conversational context and insights from analysis of human-human interactions with the design; (2) joint modeling of dialogue act selection and response generation in open domain dialogue generation; (3) proposal of a supervised learning approach and a reinforcement learning approach for model optimization; (4) empirical verification of the effectiveness of the model through automatic metrics, human annotations, machine-machine simulation, and human-machine conversation.

Table 1: Definition of dialogue acts.

| Dialogue Acts | Definitions | Examples |
|---|---|---|
| Context Maintain Statement (CM.S) | A user or a bot aims to maintain the current conversational context (e.g., topic) by giving information, suggesting something, or commenting on the previous utterances, etc. | "**there are many good places in Tokyo.**" after "I plan to have a tour in Tokyo this summer.". |
| Context Maintain Question (CM.Q) | A user or a bot asks a question in the current context. Questions cover 5W1H and yes-no with various functions such as context clarification, confirmation, knowledge acquisition, and rhetorical questions, etc. | "**where are you going to stay in Tokyo?**" after "I plan to have a tour in Tokyo this summer.". |
| Context Maintain Answer (CM.A) | A response or an answer to the previous utterances in the current context. | "**this summer.**" after "when are you going to Tokyo?". |
| Context Switch Statement (CS.S) | Similar to CM.S, but the user or the bot tries to switch to a new context (e.g., topic) by bringing in new content. | "**I plan to study English this summer.**" after "I plan to have a tour in Tokyo this summer.". |
| Context Switch Question (CS.Q) | A user or a bot tries to change the context of conversation by asking a question. | "**When will your summer vacation start?**" after "I plan to have a tour in Tokyo this summer." |
| Context Switch Answer (CS.A) | The utterance not only replies to the previous turn, but also starts a new topic. | "**I don't know because I have to get an A+ in my math exam.**" after "when are you going to Tokyo?". |
| Others (O) | greetings, thanks, and requests, etc.. | "**thanks for your help.**" |

## 2 DIALOGUE ACTS FOR OPEN DOMAIN CONVERSATION

We first define dialogue acts, and then describe the data for learning and the insights we obtain from the data. Finally, we elaborate how we build the classifier with neural networks.

### 2.1 DEFINITION OF DIALOGUE ACTS

Our dialogue acts are inherited from the existing work on 1-on-1 live chats and twitter (Kim et al., 2010; Ivanovic, 2005). Similar to (Oraby et al., 2017), we organize the 12 acts in (Ivanovic, 2005) which originate from the 42 tags (Jurafsky et al., 1997; Stolcke et al., 2006) based on the DAMSL

annotation scheme (Core & Allen, 1997) into high-level dialogue acts: "statement" and "expressive" are merged as "statement"; "yes-no question" and "open question" are combined as "question"; "yes-answer", "no-answer", and "response-ack" are collapsed as "answer"; and other tags are treated as "others". On top of these acts, we further define two high-level dialogue acts that describe how people behave regarding to conversational context in their interactions. As will be seen later, the extension may bring us further insights on engagement in social chat. Details of the dialogue acts are described in Table 1.

The high-level dialogue acts in Table 1 are generally applicable to open domain dialogues from various sources in different languages such as Twitter, Reddit, Facebook, Weibo (`www.weibo.com`), and Baidu Tieba (`https://tieba.baidu.com/`), etc. One can extend the taxonomy by defining finer-grained dialogue acts and learn their generation models with the approaches described later. Existing annotated data sets (e.g., the Switchboard Corpus[1]) do not have dialogue acts regarding to conversational context. Therefore, it is not clear how such dialogue acts depict human behavior in interactions, and there are no large scale data available for learning dialgoue generation with the dialogue acts either. To resolve these problems, we build a data set.

## 2.2 Data Set

We crawled 30 million dyadic dialogues (conversations between two people) from Baidu Tieba. Baidu Tieba is the largest Reddit-like forum in China which allows users to post and comment on others' post. Two people can communicate with each other through one posting a comment and the other one replying to the comment. Data in Baidu Tieba covers a large variety of topics, and thus can be viewed as a simulation of open domain conversation in a chatbot. We randomly sample 9 million dialogues as a training set, 90 thousand dialogues as a validation set, and 1000 dialogues as a test set. These data are used to learn a dialogue generation model later. We employ the Standford Chinese word segmenter[2] to tokenize utterances in the data. Table 2 reports statistics of the data.

Table 2: Statistics of the experimental data sets

|                             | train  | val    | test   |
|-----------------------------|--------|--------|--------|
| # dialogues                 | 9M     | 90k    | 1000   |
| Min. # turns per dialogue   | 3      | 5      | 5      |
| Max. # turns per dialogue   | 50     | 50     | 50     |
| Avg. # turns per dialogue   | 7.68   | 7.67   | 7.66   |
| Avg. # words per utterance  | 15.81  | 15.89  | 15.74  |

For dialogue act learning, we randomly sample 500 dialogues from the training set and recruit 3 native speakers to label dialogue acts[3] for each utterance according to the definitions in Table 1. Table 3 shows a labeling example from one annotator. Each utterance receives 3 labels, and the Fleiss' kappa of the labeling work is 0.45, indicating moderate agreement among the labelers.

Table 3: An example of dialogue with labeled acts.

| Turns | Dialogue Acts |
|-------|---------------|
| A: 万里长城很漂亮！The Great Wall of China is beautiful! | CM.S |
| B: 你在长城看日落了吗? Did you see the sunset on the Great Wall? | CM.Q |
| A: 是的，那是最漂亮的景色。Yes, it's the most beautiful scenery. | CM.A |
| B: 上次我去的时候人很多。It was very crowded when I visited there last time | CS.S |
| A: 我只待了一小会儿，人太多了！I only stayed there for a while. Too many vistors! | CM.S |

---

[1] `https://github.com/cgpotts/swda`

[2] `https://nlp.stanford.edu/software/tokenizer.shtml`

[3] By default, the first utterance in a dialogue is labeled as CM.* except clear opening expressions such as "hello" or "morning", because the first utterance often follows the context of a post.

### 2.3 INSIGHTS FROM THE LABELED DATA

The frequencies of the dialogue acts in terms of percentages of the total number of utterances in the labeled data are CM.S $55.8\%$, CM.Q $11.7\%$, CM.A $12.2\%$, CS.S $12.4\%$, CS.Q $4.8\%$, CS.A $2\%$, and O $1.1\%$. In addition to the numbers, we also get further insights from the data that are instructive to our dialogue generation learning:

**Context switch** is a common skill to keep conversation going. In fact, we find that $78.2\%$ dialogues contain at least one CS.* act. The average number of turns of dialogues that contain at least one CS.* is $8.4$, while the average number of turns of dialogues that do not contain a CS.* is 7. When dialogues are shorter than 5 turns, only $47\%$ of them contain a CS.*, but when dialogues exceed 10 turns, more than $85\%$ of them contain a CS.*. Because there are no specific goals in their conversations, people seldom stay long in one context. The average number of turns before context switch is $3.39$. We also observed consecutive context switch in many dialogues ($43.7\%$). The numbers suggest dialogue generation with smooth context switch and moderate context maintenance.

**Question** is an important building block in open domain conversation. In fact, $13.9\%$ CM.* are CM.Q and the percentage is even higher in CS.* which is $20.27\%$. People need to ask questions in order to maintain contexts. The average number of turns of contexts with questions (i.e., consecutive CM.* with at least one CM.Q) is $3.92$, while the average number of turns of contexts without questions is only $2.95$. The observation indicates that a good dialogue model should be capable of asking questions properly, as suggested by Li et al. (2017a). A further step to study human's questioning behavior is to look into types and functions of questions. We leave it as future work.

The observations raise new challenges that are difficult for the existing end-to-end methods to tackle (e.g., smoothly interleaving context blocks with switch actions), and thus encourage us to create a new model. Note that these observations may relate to dialogue scenarios (e.g., chatting online instead of face-to-face) and cultures, but we ignore these factors and just study how to learn the conversational patterns from the data with a principled approach. The learning approach is generally applicable to other data. To perform learning, we need to build a classifier that can automatically tag large scale dialogues with dialogue acts.

### 2.4 DIALOGUE ACT CLASSIFICATION

We aim to learn a classifier $c$ from $\mathcal{D}_A = \{d_i\}_{i=1}^N$ where $d_i = \{(u_{i,1}, a_{i,1}), \ldots, (u_{i,n_i}, a_{i,n_i})\}$ represents a dialogue with $u_{i,k}$ the $k$-th utterance and $a_{i,k}$ the labeled dialogue act. Given a new dialogue $d = \{u_1, \ldots, u_n\}$, $c$ can sequentially tag the utterances in $d$ with dialogue acts by taking $u_i$, $u_{i-1}$, and the predicted $a_{i-1}$ as inputs and outputting a vector $c(u_i, u_{i-1}, a_{i-1})$ where the $j$-th element representing the probability of $u_i$ being tagged as the $j$-th dialogue act.

We parameterize $c(\cdot, \cdot, \cdot)$ using neural networks. Specifically, $u_i$ and $u_{i-1}$ are first processed by bidirectional recurrent neural networks with gated recurrent units (biGRUs) (Chung et al., 2014) respectively. Then the last hidden states of the two biGRUs are concatenated with an embedding of $a_{i-1}$ and fed to a multi-layer perceptron (MLP) to calculate a dialogue act distribution. Formally, suppose that $u_i = (w_{i,1}, \ldots, w_{i,n})$ where $w_{i,j}$ is the embedding of the $j$-th word, then the $j$-th hidden state of the biGRU is given by $h_{i,j} = [\overrightarrow{h}_{i,j}; \overleftarrow{h}_{i,j}]$ where $\overrightarrow{h}_{i,j}$ is the $j$-th state of a forward GRU, $\overleftarrow{h}_{i,j}$ is the $j$-th state of a backward GRU, and $[\cdot; \cdot]$ is a concatenation operator. $\overrightarrow{h}_{i,j}$ and $\overleftarrow{h}_{i,j}$ are calculated by

$$\overrightarrow{h}_{i,j} = f_{\text{GRU}}(\overrightarrow{h}_{i,j-1}, w_{i,j}); \; \overleftarrow{h}_{i,j} = f_{\text{GRU}}(\overleftarrow{h}_{i,j+1}, w_{i,j}). \tag{1}$$

Similarly, we have $h_{i-1,j}$ as the $j$-th hidden state of $u_{i-1}$. Let $e(a_{i-1})$ be the embedding of $a_{i-1}$, then $c(u_i, u_{i-1}, a_{i-1})$ is defined by a two-layer MLP:

$$c(u_i, u_{i-1}, a_{i-1}) = f_{\text{MLP}}([h_{i,n}; h_{i-1,n}; e(a_{i-1})]), \tag{2}$$

where we pad zeros for $u_0$ and $a_0$ in $c(u_1, u_0, a_0)$. We learn $c(\cdot, \cdot, \cdot)$ by minimizing cross entropy with $\mathcal{D}_A$. Let $p_j(a_i)$ be the probability of $a_i$ being the $j$-th dialogue act and $c(u_i, u_{i-1}, a_{i-1})[j]$ be the $j$-th element of $c(u_i, u_{i-1}, a_{i-1})$, then the objective function of learning is formulated as

$$-\sum_{i=1}^{N} \sum_{k=1}^{n_i} \sum_{j=1}^{7} p_j(a_{i,k}) \log(c(u_{i,k}, u_{i,k-1}, a_{i,k-1})[j]). \tag{3}$$

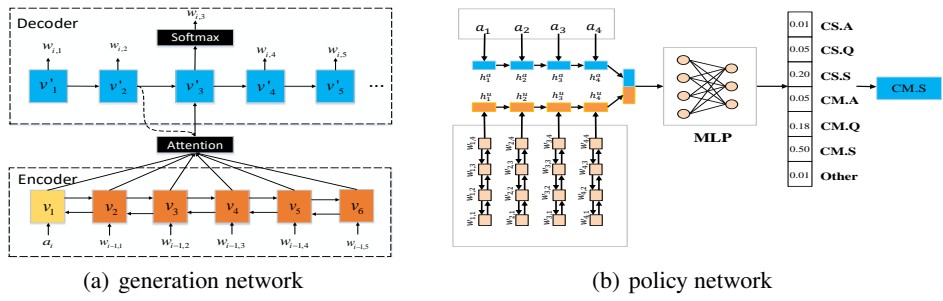

(a) generation network           (b) policy network

Figure 1: Policy network and generation network.

We randomly split the labeled dialogues as 400/30/70 dialogues with 3280/210/586 utterances for training/validation/test. Details of learning are given in Appendix 7.2. The learned classifier achieves an accuracy of 70.1% on the test data. We employ it to tag the training, validation, and test sets in Table 2.

## 3 DIALOGUE GENERATION MODEL

We present dialogue generation learning using large scale dialogues tagged with dialogue acts. Then, we describe model optimization with reinforcement learning for long-term conversation.

### 3.1 SUPERVISED LEARNING

We aim to learn a dialogue generation model $g$ from $\mathcal{D} = \{d_i\}_{i=1}^N$ where $d_i = \{(u_{i,1}, a_{i,1}), \ldots, (u_{i,n_i}, a_{i,n_i})\}$ refers to a human-human dialogue with $u_{i,k}$ the $k$-th utterance and $a_{i,k}$ the dialogue act tagged by the classifier in Section 2.4. Given $s_i = \{(u_1, a_1), \ldots, (u_{i-1}, a_{i-1})\}$ as a new dialogue session, $g(s_i)$ can generate a response as the next turn of the dialogue.

Our dialogue model consists of a policy network and a generation network. A dialogue act is first selected from the policy network according to the conversation history, and then a response is generated from the generation network based on the conversation history and the dialogue act. Formally, the dialogue model can be formulated as

$$g(s_i) = p_r(r_i|s_i, a_i^\star), \tag{4}$$

where $a_i^\star = O(p_a(a_i|s_i))$ is the selected dialogue act for the $i$-th turn, and $r_i$ is the response of the $i$-th turn. $p_a$ is the policy network and $p_r$ is the generation network. $O(\cdot)$ refers to a dialogue act select operation according to the value of the policy network. A simple defintion of $O(p_a(a_i|s_i))$ is

$$O(p_a(a_i|s_i)) = \arg\max_{a_i \in \mathbb{A}} p_a(a_i|s_i), \tag{5}$$

where $\mathbb{A}$ is the space of dialogue acts. One can also customize $O(\cdot)$ with more complicated rules to achieve controllability or further optimization (e.g., improving response diveristy by selecting multiple acts) of their systems.

Figure 1(b) shows the architecture of the policy network. The utterance sequence and the act sequence are encoded with a hierarchical encoder and a GRU encoder respectively. Then, the last hidden states of the two encoders are concatenated and fed to an MLP to calculate a probability distribution of dialogue acts for the next turn. Formally, $\forall u_j \in s_i$, $u_j$ is first transformed to hidden vectors $\{h_{j,k}^u\}_{k=1}^{n_j'}$ through a biGRU parameterized as Equation (1). Then, $\{h_{j,n_j'}^u\}_{j=1}^{i-1}$ is processed by a GRU parameterized as $t_k = f_{\text{GRU}}^u(t_{k-1}, h_{k,n_k'}^u)$. In parallel, $\{a_1, \ldots, a_{i-1}\}$ is transformed to $\{h_k^a\}_{k=1}^{i-1}$ by $h_k^a = f_{\text{GRU}}^a(h_{k-1}^a, e(a_k))$. $p_a(a_i|s_i)$ is then defined by

$$p_a(a_i|s_i) = f_{\text{MLP}}([t_{i-1}; h_{i-1}^a]). \tag{6}$$

We build the generation network in a sequence-to-sequence framework. Here, we simplify $p_r(r_i|s_i, a_i)$ as $p_r(r_i|a_i, u_{i-1}, u_{i-2})$ since decoding natural language responses from long conversation history is challenging. Figure 1(a) illustrates the architecture of the generation network. The

only difference from the standard encoder-decoder architecture with an attention mechanism is that in encoding, we concatenate $u_{i-1}$ and $u_{i-2}$, and attach $a_i$ to the top of the long sentence as a special word. The technique here is similar to that in zero-shot machine translation (Johnson et al., 2016). More formulation details can be found in Appendix 7.1.

The dialogue model is then learned by minimizing the negative log likelihood of $\mathcal{D}$:

$$-\sum_{i=1}^{N}\sum_{k=1}^{n_i}[\log(p_r(u_{i,k}|d_{i,<k},a_{i,k})) + \log(p_a(a_{i,k}|d_{i,<k}))], \tag{7}$$

where $d_{i,<k} = \{(u_{i,1},a_{i,1}),\ldots,(u_{i,k-1},a_{i,k-1})\}$. Through supervised learning, we fit the dialogue model to human-human interactions in order to learn their conversational patterns. However, supervised learning does not explicitly encourage long-term conversation (e.g., 45.35% dialogues in our training set are no more than 5 turns). The policy network is learned by fitting to the existing conversation history, and it is not aware what is going to happen in the future when a dialogue act is selected. This motivates us to further optimize the model through a reinforcement learning approach.

## 3.2 Reinforcement Learning

We aim to optimize the dialogue model by letting it know a possible result in the following conversation when an act and a response are generated. To avoid exhausting and expensive online optimization, we choose self-play (Li et al., 2016b; Lewis et al., 2017) where we let two models learned with the supervised approach talk to each other in order to improve their performance. In the simulation, a dialogue is initialized with a message sampled from the training set. Then, the two models continue the dialogue by alternately taking the conversation history as an input and generating a response (top one in beam search) until $T$ turns ($T = 20$ in our experiments).

To speed up training and avoid generated responses diverging from human language, we fix the generation network and only optimize the policy network by reinforcement learning. Thus, the policy in learning is naturally defined by the policy network $p_a(a_i|s_i)$ with $s_i = \{(u_1,a_1),\ldots,(u_{i-1},a_{i-1})\}$ a state and $a_i$ an action. We define a reward function $r(a_i,s_i)$ as

$$r(a_i,s_i) = \alpha\mathbb{E}[len(a_i,s_i)] + \beta\mathbb{E}[rel(a_i,s_i)], \tag{8}$$

where $\mathbb{E}[len(a_i,s_i)]$ is the expected conversation length after taking action $a_i$ under state $s_i$, $\mathbb{E}[rel(a_i,s_i)]$ is the expected response relevance within the conversation, $\alpha = 0.67$, and $\beta = 0.33$. Through Equation (8), we try to encourage actions that can lead to long (measured by $\mathbb{E}[len(a_i,s_i)]$) and reasonable (measured by $\mathbb{E}[rel(a_i,s_i)]$) conversations.

To estimate $\mathbb{E}[len(a_i,s_i)]$ and $\mathbb{E}[rel(a_i,s_i)]$, we fix $(s_i,a_i)$ and construct a dialogue set $\{d'_{i,j}\}_{j=1}^{N}$ ($N = 10$ in our experiments) by sampling after $(s_i,a_i)$ with self-play. $\forall j$, $d'_{i,j} = (s_i, u_{i,i+1},\ldots,u_{j,n_{i,j}})$ where $\forall k$, $u_{j,i+k}$ is randomly sampled from the top 5 beam search results of $p_r$ conditioned on the most probable dialogue act given by $p_a$ for that turn. Inspired by (Li et al., 2016b), we terminate a simulated dialogue if (1) $cosine(e(u_{i-1}),e(u_i)) > 0.9$ && $cosine(e(u_i)),e(u_{i+1})) > 0.9$, or (2) $cosine(e(u_{i-1}),e(u_{i+1})) > 0.9$, or (3) the length of the dialogue reaches $T$, where $e(\cdot)$ denotes the representation of an utterance given by the encoder of $p_r$. Condition (1) means three consecutive turns are (semantically) repetitive, and Condition (2) means one agent gives repetitive responses in two consecutive turns. Both conditions indicate a high probability that the conversation falls into a bad infinite loop. $\mathbb{E}[len(a_i,s_i)]$ and $\mathbb{E}[rel(a_i,s_i)]$ are then estimated by

$$\mathbb{E}[len(a_i,s_i)] = \frac{1}{N}\sum_{j=1}^{N}n_{i,j}; \ \mathbb{E}[rel(a_i,s_i)] = \frac{1}{N}\sum_{j=1}^{N}\frac{1}{n_{i,j}}\sum_{k=1}^{n_{i,j}}m(d_{i,j<k},u_{j,k}), \tag{9}$$

where $d_{i,j<k} = (u_1,\ldots,u_{j,k-1})$, and $m(\cdot,\cdot)$ is the dual LSTM model proposed in (Lowe et al., 2015) which measures the relevance between a response and a context. We train $m(\cdot,\cdot)$ with the 30 million crawled data through negative sampling. The objective of learning is to maximize the expected future reward:

$$\mathcal{J}(\theta) = \mathbb{E}[\sum_{i=1}^{T}r(a_i,s_i)]. \tag{10}$$

The gradient of the objective is calculated by Reinforce algorithm (Williams, 1992):

$$\partial_\theta \mathcal{J} \approx \sum_{t=1}^{T} \partial_\theta \log(p_a(a_t|s_t)) \Big( \sum_{i=t}^{T} (r(a_i, s_i) - b_t) \Big), \tag{11}$$

where the baseline $b_t$ is empirically set as $\frac{1}{|\mathbb{A}|} \sum_{a_t \in \mathbb{A}} r(a_t, s_t)$.

## 4 EXPERIMENT

### 4.1 EXPERIMENT SETUP

Our experiments are conducted with the data in Table 2. The following methods are employed as baselines: (1) **S2SA**: sequence-to-sequence with attention (Bahdanau et al., 2015) in which utterances in contexts are concatenated as a long sentence. We use the implementation with Blocks (`https://github.com/mila-udem/blocks`); (2) **HRED**: the hierarchical encoder-decoder model in (Serban et al., 2016) implemented with the source code available at (`https://github.com/julianser/hed-dlg-truncated`); (3) **VHRED**: the hierarchical latent variable encoder-decoder model in (Serban et al., 2017b) implemented with the source code available at (`https://github.com/julianser/hed-dlg-truncated`); and (4) **RL-S2S**: dialogue generation with reinforcement learning (Li et al., 2016b). We implement the algorihtm by finishing the code at (`https://github.com/liuyuemaicha/Deep-Reinforcement-Learning-for-Dialogue-Generation-in-tensorflow`). Dull responses are defined as in (Li et al., 2016b) and listed in Appendix 7.3.

All baseline models are implemented with the recommended configurations in the existing literatures. We denote our Dialogue Act aware Generation Model with only Supervised Learning as SL-DAGM, and the full model (supervised learning + reinforcement learning) as RL-DAGM. Implementation details are given in Appendix 7.3.

### 4.2 RESPONSE GENERATION FOR GIVEN CONTEXTS

The first experiment is to check if the proposed models can generate high-quality responses regarding to given contexts. To this end, we take the last turn of each test dialogue as ground truth, and feed the previous turns as a context to different models for response generation. Top one responses from beam search (beam size= 20) of different models are collected, randomly shuffled, and presented to 3 native speakers to judge their quality. Each response is rated by the three annotators under the following criteria: **2**: the response is not only relevant and natural, but also informative and interesting; **1**: the response can be used as a reply, but might not be informative enough (e.g.,"Yes, I see" etc.); **0**: the response makes no sense, is irrelevant, or is grammatically broken.

Table 4: Evaluation Results

(a) Human annotations. Ratios are calculated by combining labels from the three judges.

| | 0 | 1 | 2 | Kappa |
|---|---|---|---|---|
| S2SA | 0.478 | 0.478 | 0.044 | 0.528 |
| HRED | 0.447 | 0.456 | 0.097 | 0.492 |
| VHRED | 0.349 | 0.471 | 0.180 | 0.494 |
| RL-S2S | 0.393 | 0.462 | 0.142 | 0.501 |
| SL-DAGM | 0.279 | 0.475 | 0.244 | 0.508 |
| RL-DAGM | 0.341 | 0.386 | 0.273 | 0.485 |

(b) Average dialogue length in machine-machine and human-machine conversations.

| | Machine-Machine | Human-Machine |
|---|---|---|
| RL-S2S | 4.36 | 4.54 |
| SL-DAGM | 7.36 | 5.24 |
| RL-DAGM | 7.87 | 5.58 |

Table 4(a) summarizes the annotation results. Improvements from our models over the baseline methods are statistically significant (t-test, p-value $< 0.01$). In addition to human annotations, we also compare different models using automatic metrics with the the ground truth. These metrics include (1) BLEU (Papineni et al., 2002) which measures term overlap of two responses; (2) embedding based metrics (Liu et al., 2016) such as Embedding Average (Average), Embedding Extrema (Extrema), and Embedding Greedy (Greedy) which measure similarity of two responses in a semantic space; and (3) ratios of distinct unigrams (distinct-1) and bigrams (distinct-2) in the generated responses which are employed in (Li et al., 2015) to measure response diversity. Table 5 reports the automatic evaluation results.

Table 5: Automatic evaluation results. Numbers in bold mean that improvement from the model on that metric is statistically significant over the baseline methods (t-test, p-value $< 0.01$).

| | BLEU-1 | BLEU-2 | Average | Extrema | Greedy | Distinct-1 | Distinct-2 |
|---|---|---|---|---|---|---|---|
| S2SA | 4.67 | 1.18 | 21.45 | 16.68 | 21.53 | 0.033 | 0.069 |
| HRED | 3.70 | 1.06 | 16.87 | 13.58 | 20.15 | 0.062 | 0.139 |
| VHRED | 6.10 | 1.76 | 20.83 | 16.17 | 21.36 | 0.079 | 0.225 |
| RL-S2S | 5.57 | 1.83 | 20.72 | 16.73 | 20.64 | 0.100 | 0.213 |
| SL-DAGM | 6.23 | 2.07 | 20.68 | 16.42 | 21.52 | **0.200** | **0.466** |
| RL-DAGM | 6.77 | 2.12 | 21.18 | 16.97 | 21.76 | **0.223** | **0.503** |

Table 6: An example of response generation. Utterances in the context are split by "⇒".

| Context | Responses |
|---|---|
| 一起吃晚餐？⇒ 中饭好吗？⇒ 中饭只能在公司吃。⇒ 那我不能来了。我在休假。

Have dinner together? ⇒ how about lunch? ⇒ I can only have my lunch at company. ⇒ Then I cannot join you because I am in my vacation. | **S2SA**：我也是这么想的。I think so
**HRED**：放假了啊？You are already in vacation?
**VHRED**：哈哈哈。Haha.
**RL-S2S**：我已经在吃了。I am having lunch now.
**SL-DAGM**：好吧，我刚从厦门回来，想在食堂吃碗面。OK. I am just back from Xiamen, and want to have noodle in cafeteria. (**CS.S**)
**RL-DAGM**：放假有什么安排。What are you going to do for your vacation? (**CS.Q**)
**CM.Q**：放假了吗？You are already in vacation?
**CM.S**：我还以为你没休假呢。I thought you were at work. |

We can see that one benefit brought by the dialogue acts is that diversity of responses is significantly improved. This is supported by the much more 2 responses from the two models in Table 4(a) and the significant improvement on distinct n-grams in Table 5. The reason is easy to understand: we search a response not only from a language space, but also from an act space. The dimension of dialogue acts provides further variations to the generated responses. On the other hand, due to the diversity, responses from our models may diverge from the ground truth sometimes. This is why improvements on other automatic metrics are not significant. To further explain the advantages of our models, we show an example in Table 6. In addition to responses from SL-DAGM and RL-DAGM which are selected from the dialogue acts obtained by Equation (5), we also show responses from other reasonable but not selected acts. With dialogue acts, responses from our models become really rich, from confirmation (CM.Q) to an open question (CS.Q) and then to a long informative statement (CS.S). More importantly, the dialogue acts let us know why we have such responses: both SL-DAGM and RL-DAGM try to switch to new topics (e.g., Xiamen, noodle, and plan etc.) in order to continue the conversation. One can also change the flow of the conversation by picking responses from other dialogue acts. The example demonstrates that in addition to good performance, our models enjoy good interpretability and controllability as well. We show more such examples in Appendix 7.4.

### 4.3 ENGAGEMENT TEST

Secondly, we study conversation engagement with the proposed models. Experiments are conducted through machine-machine simulation and human-machine conversation. In both experiments, we compare SL-DAGM and RL-DAGM with RL-S2S, as RL-S2S is the only baseline optimized for future success. Responses from all models are randomly sampled from the top 5 beam search results. Average length of dialogues is employed as an evaluation metric.

Machine-machine simulation is conducted in a way similar to (Li et al., 2016b) in which we let two bots equipped with the same model talk with each other in 1000 simulated dialogues. Each dialogue is initialized with the first utterance of a test example, and terminated according to the termination conditions for reward estimation in Section 3.2. In human-machine conversation, we recruit 5 native speakers as testers and ask them to talk with the bots equipped with the three models. Every time, a bot is randomly picked for a tester, and the tester does not know which model is behind. Every tester finishes 100 dialogues with each bot. To make a fair comparison, we let the bots start dialgoues. A starting message in a dialogue is randomly sampled from the test data and copied 3

times for all the 3 bots (a tester can skip the message if he/she cannot understand it). A dialogue is terminated if (1) the tester thinks the conversation cannot be continued (e.g., due to bad relevance or repetitive content etc.); or (2) the bot gives repetitive responses in two consecutive turns (measured by $cosine(e(u_{i-1}), e(u_{i+1})) > 0.9$). Dialogue acts in human turns are tagged by the classifier in Section 2.4. The evaluation metric is calculated with the total $500$ dialogues for each model.

Table 4(b) reports the evaluation results. In both experiments, SL-DAGM and RL-DAGM can lead to longer conversations, and the improvements from both models over the baseline are statistically significant (t-test, p-value $< 0.01$). Improvements in human-machine conversation are smaller than those in machine-machine simulation, indicating the gap between the simulation environment and the real conversation environment and encouraging us to consider online optimization in human-machine conversations in the future. RL-DAGM is better than SL-DAGM in both experiments, indicating the efficacy of reinforcement learning.

The reason that our models are better is that they captured conversational patterns in human-human interactions and obtained further optimization through reinforcement learning. First, the models can pro-actively switch contexts in a smooth way. In machine-machine simulation, $65.4\%$ (SL) and $94.4\%$ (RL) dialogues contain at least one CS.*; and in human-machine conversation, the two percentages are $38.1\%$ (SL) and $48.1\%$ (RL) respectively. More interestingly, in machine-machine simulation, average lengths of dialogues without CS.* are only $4.78$ (SL) and $2.67$ (RL) respectively which are comparable with or even worse than RL-S2S, while average lengths of dialogues with CS.* are $8.66$ (SL) and $8.18$ (RL) respectively. The results demonstrate the importance of context switch for engagement in open domain conversation and one signficant effect of RL is promoting context switch in interactions for future engagment even with a little sacrifice on relevance of the current turn (e.g., more 0 responses than SL-DAGM in Table 4(a)). Second, the models can drive conversations by asking questions. In machine-machine simulation, $36.5\%$ (SL) and $32.4\%$ (RL) dialogues contain at least one question. The percentages in human-machine conversation are $17.7\%$ (SL) and $22.3\%$ (RL) respectively. We give more analysis in Appendix 7.5.

## 4.4 DISCUSSION

Finally, we study how the generated responses are affected by the dialogue acts. We collect generated responses from a specific dialogue act for the contexts of the test dialogues, and characterize the responses with the following metrics: (1) distinct-1 and distinct-2; (2) words out of context (OOC): ratio of words that are in the generated responses but not contained by the contexts; and (3) average length of the generated responses (Ave Len).

Table 7: Characteristics of the generated responses from different dialogue acts.

|  | Distinct-1 | Distinct-2 | OOC | Ave Len |
|---|---|---|---|---|
| CM.S | 0.114 | 0.262 | 0.091 | 5.57 |
| CM.Q | 0.092 | 0.220 | 0.038 | 5.21 |
| CM.A | 0.119 | 0.269 | 0.094 | 5.58 |
| CS.S | 0.250 | 0.521 | 0.168 | 8.21 |
| CS.Q | 0.223 | 0.460 | 0.152 | 5.85 |
| CS.A | 0.244 | 0.500 | 0.166 | 8.42 |

Table 7 reports the results[4]. In general, responses generated from CS.* are longer, more informative, and contain more new words than responses generated from CM.*, which has been illustrated by the example in Table 6. Another interesting finding is that statements and answers are generally more informative than questions in both CS.* and CM.*. In addition to these metrics, we also calculate BLEU scores and embedding based metrics, but do not observe significant difference among responses from different dialogue acts. The reason might be that these metrics are based on comparsion of the generated responses and human responses, but human responses in the test set are inherently mixture of responses from different dialogue acts.

---

[4]We omit the dialogue act O, as only $1.1\%$ of the labeled data are O and it is difficult for neural networks to capture the characteristics of text from such a few data. In practice, one can generate text for O by editorial.

## 5 RELATED WORK

Existing dialogue models are either built for open domain conversation or for specific task completion. Regarding to the former, a common practice is to learn a generation model in an end-to-end fashion. On top of the basic sequence-to-sequence with attention architecture (Vinyals & Le, 2015; Shang et al., 2015), various extensions have been proposed to tackle the "safe response" problem (Li et al., 2015; Mou et al., 2016; Xing et al., 2017); to model complicated structures of conversation contexts (Serban et al., 2016; Sordoni et al., 2015); to bias responses to some specific persona or emotions (Li et al., 2016a; Zhou et al., 2017); and to pursue better optimization strategies (Li et al., 2017b; 2016b). On the other line of research, POMDP (Young et al., 2013) breaks down the development of task-oriented dialogue systems into natural language understanding (Yao et al., 2014; Henderson et al., 2014), dialogue management (Mrkšić et al., 2016), and response generation (Wen et al., 2015). Recently, researchers also consider learning task-oriented dialogue models in an end-to-end way (Wen et al., 2016; 2017; Bordes & Weston, 2017). In this work, we introduce dialogue acts into open domain dialogue generation. Although some previous work (Zhao et al., 2017; Serban et al., 2017a) has leveraged dialogue acts as extra features, the dialogue acts in this work are generally designed for explaining engagement in social chat and modelled as policies to manage the flow of interactions. To the best of our knowledge, we are the first who design dialogue acts to explain social interactions, control open domain response generation, and guide human-machine conversations.

Before us, some researchers have proposed analyzing open domain dialogues with dialogue acts (Kim et al., 2010; 2012; Oraby et al., 2017; Ivanovic, 2005; Wallace et al., 2013; Wu et al., 2005; Ritter et al., 2010). These work, however, stops at performing utterance classification or clustering. Our dialogue act design is inspired by these work, but we not only exploit the dialogue acts to interpret open domain dialogues, but also conduct dialogue generation with the dialogue acts.

## 6 CONCLUSION

We study open domain dialogue generation with generally designed dialogue acts that can describe human behavior in social interactions. To mimic such behavior, we propose jointly modeling dialogue act selection and response generation, and perform both supervised learning with a learned dialogue act classifier and reinforcement learning for long-term conversation. Empirical studies on response generation for given contexts, machine-machine simulation, and human-machine conversation show that the proposed models can significantly outperform state-of-the-art methods.

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

## 7 APPENDIX

### 7.1 GENERATION NETWORK

Suppose that $x_i = [a_i; u_{i-1}; u_{i-2}] = (w_{i,1}, \ldots, w_{i,n_i'})$ where $w_{i,k}$ is the embedding of the $k$-th word, then the $k$-th hidden state of the encoder is given by $v_{i,k} = [\overrightarrow{v}_{i,k}; \overleftarrow{v}_{i,k}]$ where

$$\overrightarrow{v}_{i,k} = f_{\text{GRU}}^e(\overrightarrow{v}_{i,k-1}, w_{i,k}); \overleftarrow{v}_{i,k} = f_{\text{GRU}}^e(\overleftarrow{v}_{i,k+1}, w_{i,k}) \tag{12}$$

Positions of $u_{-1}$ and $u_0$ in $x_1$ and $x_2$ are padded with zeros. Let $r_i = (w_{i,1}', \ldots, w_{i,T}')$, then in decoding the $j$-th word $w_{i,j}'$, $\{v_{i,1}, \ldots, v_{i,n_i'}\}$ is summarized as a context vector $c_{i,j}$ through an attention mechanism:

$$c_{i,j} = \sum_{k=1}^{n_i'} \alpha_{j,k} v_{i,k}; \ \alpha_{j,k} = \frac{exp(e_{j,k})}{\sum_{m=1}^{n_i'} exp(e_{j,m})}; \ e_{j,k} = v^\top tanh(W_\alpha[v_{i,k}; v_{i,j-1}']), \tag{13}$$

where $v$ and $W_\alpha$ are parameters, and $v_{i,j-1}'$ is the $(j-1)$-th hidden state of the decoder GRU in which $v_{i,j}'$ is calculated by

$$v_{i,j}' = f_{\text{GRU}}^d(v_{i,j-1}', w_{i,j-1}', c_{i,j}). \tag{14}$$

The generation probability of $w_{i,j}$ is then defined as

$$p_r(w_{i,j}'|w_{i,<j}', x_i) = \mathcal{I}(w_{i,j}')^\top softmax(w_{i,j-1}', v_{i,j}'), \tag{15}$$

where $\mathcal{I}(w_{i,j}')$ is a vector with only one element 1 indicating the index of $w_{i,j}'$ in the vocabulary. $p_r(r_i|a_i, u_{i-1}, u_{i-2})$ is finally defined as

$$p_r(r_i|a_i, u_{i-1}, u_{i-2}) = p_r(w_{i,1}'|x_i) \prod_{j=2}^{T} p_r(w_{i,j}'|w_{i,<j}', x_i). \tag{16}$$

### 7.2 DETAILS OF LEARNING THE DIALOGUE ACT CLASSIFIER

We randomly split the 500 labeled dialogues as 400, 30, and 70 dialogues for training, validation, and test respectively. Utterances in the three sets are 3280, 210, and 586 respectively. In training, we represent dialogue acts as probability distributions by averaging the labels given by the three annotators. For example, if an utterance is labeled as "CM.S", "CM.S", and "CS.S", then the probability distribution is $(0.67, 0, 0, 0.33, 0, 0, 0)$. In test, we predict the dialogue act of an utterance $u_i$ by $\arg\max_j g(u_i, u_{i-1}, a_{i-1})[j]$. To avoid overfitting, we pre-train word embeddings using word2vec[5] with an embedding size of 200 on the 30 million data and fix them in training. We set the embedding size of the dialogue acts and the hidden state size of the biGRUs as 100, and the dimensions of the first layer and the second layer of the MLP as 200 and 7 respectively. We optimize the objective function (3) using back-propagation and the parameters are updated by stochastic gradient descent with AdaDelta algorithm (Zeiler, 2012). The best performing model on the validation data is picked up for test.

### 7.3 DETAILS OF IMPLEMENTATION OF THE DIALOGUE GENERATION MODEL

In learning of the generation network, we set the size of word embedding as 620 and the size of hidden vectors as 1024 in both the encoder and the decoder. Both the encoder vocabulary and the decoder vocabulary contain $30,000$ words. Words out of the vocabularies are replaced by a special token "UNK". We employ AdaDelta algorithm (Zeiler, 2012) to train the generation network with a batch size 128. We set the initial learning rate as 1.0 and reduce it by half if perplexity on validation begins to increase. We stop training if the perplexity on validation keeps increasing in two successive epochs.

In learning of the policy network, we set the size of word embedding, the size of dialogue act, and the size of hidden states of the biGRU as 100. There are 50 neurons in the first layer of the MLP and 7 neurons in the second layer of the MLP. Vectors in the policy network have smaller sizes than

---

[5]https://code.google.com/archive/p/word2vec/

those in the generation network because the complexity of dialogue act prediction is much lower than language generation.

In reinforcement learning, the size of mini-batch is 60 and learning rate is fixed as 0.05. To estimate the reward, we train a dual LSTM (Lowe et al., 2015) with the size of word embedding and the size of hidden states as 100. Responses from the simulated dialogues are generated with a beam size 20.

In RL-S2S, we define 8 responses as dull responses according to the frequency of responses in the training set. Table 8 gives the responses.

Table 8: Dull responses for learning RL-S2S.

| No. | Chinese responses | English translations |
|-----|-------------------|----------------------|
| 1 | 我不知道 | I do not know. |
| 2 | 我觉得你说得对 | I think you are right. |
| 3 | 你是男的女的 | Are you a man or a woman? |
| 4 | 嗯我知道 | I see. |
| 5 | 我也不知道 | I do not know either. |
| 6 | 你说的对 | You are right. |
| 7 | 我也是这么想的 | I think so. |
| 8 | 好啊 | OK. |

## 7.4 MORE EXAMPLES OF RESPONSE GENERATION

We compare SL-DAGM and RL-DAGM with baseline models in terms of response quality for given contexts with more examples in Table 9.

## 7.5 FURTHER ANALYSIS ON ENGAGMENT TEST

Table 10 gives some examples on machine-machine simulation. Unlike the dialogues from RL-S2S which quickly converge to loops, dialogues from our models smoothly move forward under the management of the dialogue acts. The dialogue acts let us know why such responses are generated and make the simulated dialogues closer to human dialogues with moderate context continuation and jumping out of the contexts at proper timing. Table 11 and Table 12 show some examples from the test of human-machine conversation. We denote a machine turn as "M" and a human turn as "H". After each example, we give the reason of termination in which "EOD-H" means the dialogue is terminated by the tester and "EOD-R" means the dialogue is terminated by the repetition check with the next generated turn attached. Compared to dialogues with the baseline, dialogues with our models can go deeper with much richer content, although a side-effect is that sometimes responses from CS.* might be nonsense (e.g., the first example of SL-DAGM). This sheds light on our future direction to further improve the generation network with knowledge. In addition to the qualitative results, we also show quantitative results of human-machine conversation test in terms of different testers in Figure 2. Although there exists variance among the testers, the overall trend is consistent with the numbers in Table 4(b).

Table 9: More examples of response generation. Utterances in the context are split by "⇒".

| Context | Responses |
|---|---|
| 你眼袋略大，常熬夜? ⇒ 我大学毕业后没有熬夜过? ⇒ 早睡早起好孩子⇒ 是的

Your have dark circles under your eyes. Always stay up late? ⇒ I never stay up late after I graduate from my university. ⇒ Good boy! Early to bed, early to rise. ⇒ Yes. | **S2SA**: 我也是这么想的。I think so
**HRED**: 哈哈。Haha.
**VHRED**：你是在夸我吗? So you are praising me?
**RL-S2S**：我也是好孩子。I am a good boy too.
**SL-DAGM**：我经常倒时差睡眠不好。But I often have poor sleep because of jet lag. (**CS.S**)
**RL-DAGM**：想问你每天几点起床。Then when do you get up everyday? (**CS.Q**)
**CM.Q**：你要睡了吗? Are you going to sleep?
**CM.S**：我也是早睡。I sleep early as well. |
| 我们一起合租怎么样? ⇒ 多少钱? ⇒ 800 ⇒ 加我微信

How about renting an apartment together? ⇒ How much? ⇒800. ⇒ Then add me as a contact in WeChat. | **S2SA**: 邮件发你了。I have sent you an email.
**HRED**：你加我微信可以吗? Could you please add me in your WeChat?
**VHRED**：加我微信。Add me in your WeChat.
**RL-S2S**：邮件发你了。I have sent you an email.
**SL-DAGM**：我想和你一起玩桌游。I want to play board games with you. (**CS.S**)
**RL-DAGM**：什么时候联系。When can I contact you? (**CM.Q**)
**CS.Q**：你在哪个区? Which district do you live in?
**CM.S**：给我微信号。Give me your WeChat ID. |

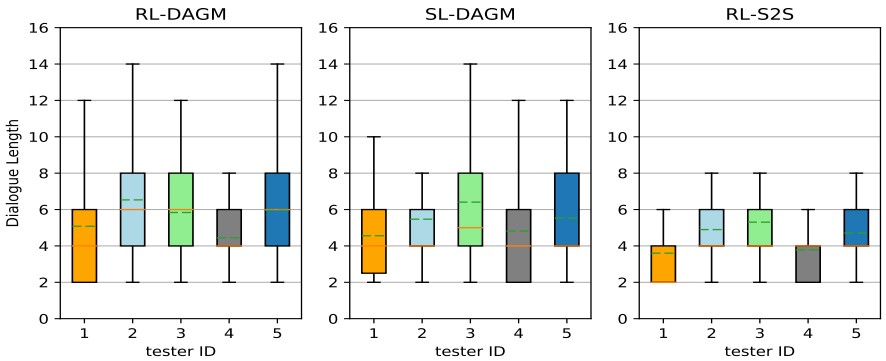

Figure 2: Evaluation results on human-machine conversation in terms of different testers.

Table 10: Comparison of simulated dialogues from different models.

| RL-S2S | SL-DAGM | RL-DAGM |
|---|---|---|
| A: 无锡妹子你好！
B: 我是无锡的，你在哪儿?
A: 我在上海。
B: 我也在上海。
A: 我也在上海。

A: Hello, Wuxi girl!
B: I come from Wuxi. How about you?
A: I come from Shanghai.
B: I come from Shanghai too.
A: I come from Shanghai too. | A: 无锡妹子你好！(CM.S)
B: 你也无锡的? (CM.Q)
A: 嗯嗯，无锡的。(CM.A)
B: 我在无锡上学呢。(CM.S)
A: 无锡有什么好吃的吗? (CS.Q)

A: Hello, Wuxi girl! (CM.S)
B: You also come from Wuxi? (CM.Q)
A: Yes, I am. (CM.A)
B: I am a college student in Wuxi. (CM.S)
A: Is there any delicious food in Wuxi? (CS.Q) | A: 无锡妹子你好！(CM.S)
B: 你好，你也在江苏? (CS.Q)
A: 不是土著，在江苏读书。(CS.S)
B: 江苏本地人很多啊。(CM.S)
A: 但是我身边都是外地人。(CM.S)

A: Hello, Wuxi girl! (CM.S)
B: Hello, you also in Jiangsu province? (Wuxi is a city in Jiangsu) (CS.Q)
A: I am not born in Jiangsu, but I go to college in Jiangsu. (CS.S)
B: Many people in Jiangsu are natives. (CM.S)
A: But most people around me are not natives. (CM.S) |
| A: 去新加坡马来西亚可以学英语.
B: 新加坡有好玩的吗?
A: 有好玩的吗?
B: 新加坡有什么好玩的? 我想去看看。

A: I can learn English in Singapore and Malaysia.
B: Are there any interesting places in Singapore?
A: Are there any interesting places?
B: Are there any interesting places in Singapore? I want to have a tour in Singapore. | A: 去新加坡马来西亚可以学英语。(CM.S)
B: 你英文名是什么? (CS.Q)
A: 我应该可以写出来。(CM.S)
B: 那你姓什么。(CM.S)
A: 我什么时候写出来? (CS.Q)

A: I can learn English in Singapore and Malaysia. (CM.S)
B: What is your English name? (CS.Q)
A: I think I can write it out. (CM.S)
B: What is your family name? (CM.S)
A: When shall I write my English name? (CS.Q) | A: 去新加坡马来西亚可以学英语。(CM.S)
B: 你也想去新加坡学英语? (CM.Q)
A: 一样一样。(CM.A)
B: 你是什么专业的? (CS.Q)
A: 我们学校有什么专业? (CM.Q)

A: I can learn English in Singapore and Malaysia. (CM.S)
B: You also want to go there to learn English? (CM.Q)
A: Yep. (CM.A)
B: What are you major in? (CS.Q)
A: What majors do we have? (CM.Q) |

Table 11: Example 1 of human-machine conversation. "M" means a machine turn, and "H" means a human turn.

| RL-S2S | SL-DAGM | RL-DAGM |
|---|---|---|
| M: 喝汤就饱了。
H: 可以减肥。
M: 减肥会不会很痛苦啊?
H: 不吃不喝确实很难受。
M: 我都不敢吃了。
H: 那就不吃了?
EOD-R (我也不敢吃了。)

M: I am full with only soup.
H: Then you can lose weight.
M: Is it tough to lose weight?
H: It's indeed tough when you have to diet.
M: I have to diet.
H: Then diet?
EOD-R (I have to diet as well.) | M: 喝汤就饱了。(CM.S)
H: 可以减肥。(CS.S)
M: 吃不胖的体质伤不起啊! (CM.S)
H: 羡慕。(CM.A)
M: 我正在增肥。(CS.S)
H: 多吃冰淇淋。(CM.S)
M: 冰与火之歌都吃过了。(CM.S)
EOD-H

M: I am full with only soup. (CM.S)
H: Then you can lose weight. (CS.S)
M: I really want to get a little fatter. (CM.S)
H: I am jealous of you! (CM.A)
M: I am trying to gain weight. (CS.S)
H: Then have more ice cream. (CM.S)
M: I even ate Song of Ice and Fire.(CM.S)
EOD-H | M: 喝汤就饱了。(CM.S)
H: 可以减肥。(CS.S)
M: 你吃蛋白粉? (CS.Q)
H: 我不吃蛋白粉。(CM.A)
M: 酵素粉是什么? (CS.Q)
H: 减肥神物。(CM.A)
M: 我去试试! (CM.S)
H: 快去吧，吃酵素要坚持。(CM.S)
M: 药店有卖酵素粉的吗? 多少钱啊? (CS.Q)
H: 有，不贵。(CM.A)
EOD-R (请问药店有卖的吗? 多少钱? )

M: I am full with only soup. (CM.S)
H: Then you can lose weight. (CS.S)
M: Do you eat albumen powder? (CS.Q)
H: Nope. (CM.A)
M: What is enzyme powder? (CS.Q)
H: Something can make you slim. (CM.A)
M: I want to try it! (CM.S)
H: Go ahead, and do not give up. (CM.S)
M: Can I get enzyme powder in pharmacy? How much is it? (CS.Q)
H: Yes you can. It's not expensive. (CM.A)
EOD-R (Can I get it in pharmacy? How much?) |

Table 12: Example 2 of human-machine conversation. "M" means a machine turn, and "H" means a human turn.

| RL-S2S | SL-DAGM | RL-DAGM |
|---|---|---|
| M: 你用淡妆比较好看。
H: 嗯我听你的。
M: 谢谢你，一起努力啊。
H: 一起努力啊。
EOD-R （一起努力！）

M: Light makeup looks good for you.
H: Well, you are right.
M: Thank you. Let's do it together!
H: Do it together!
EOD-R (Do it together!) | M: 你用淡妆比较好看。(CM.S)
H: 嗯我听你的。(CM.A)
M: 只用粉底和口红吧。(CS.S)
H: 你推荐个粉底。(CM.S)
M: 资生堂的不错。(CM.S)
H: 这个好。(CM.S)
M: 因为我用过一段时间，觉得很不错(CM.S)
H: 那太好了。(CM.S)
M: 你用过几次? (CM.Q)
H: 我没用过(CM.A)
EOD-R(你用过几次？)

M: Light makeup looks good for you. (CM.S)
H: Well, you are right. (CM.A)
M: Only use foundation and lipstick. (CS.S)
H: Recommend me a brand of foundation. (CM.S)
M: Shiseido is good. (CM.S)
H: That is great. (CM.S)
M: Because I used to try it for sometime, and feel good. (CM.S)
H: Sounds good. (CM.S)
M: How many times have you used it? (CM.Q)
H: Never. (CM.A)
EOD-R (How many times have you used it?) | M: 你用淡妆比较好看。(CM.S)
H: 嗯我听你的。(CM.A)
M: 画浓妆会不会气色很差? (CS.Q)
H: 不会的。(CM.A)
M: 那有什么副作用? (CS.Q)
H: 皮肤可能不好。(CM.S)
EOD-R (有什么副作用？)

M: Light makeup looks good for you. (CM.S)
H: Well, you are right. (CM.A)
M: Does heavy makeup make your complexion look bad? (CS.Q)
H: Nope. (CM.S)
M: Then is there any side-effect? (CS.Q)
H: My skin will become worse. (CM.S)
EOD-R (is there any side-effect?) |

