# OpenReview forum: "Towards Interpretable Chit-chat: Open Domain Dialogue Generation with Dialogue Acts"
_ICLR.cc/2018/Conference — Reject_

### Official Review · AnonReviewer2 · 2017-11-27
**Strong results from a simple idea**

**Rating:** 7
**Confidence:** 3

**Review:**

The authors use a distant supervision technique to add dialogue act tags as a conditioning factor for generating responses in open-domain dialogues. In their evaluations, this approach, and one that additionally uses policy gradient RL with discourse-level objectives to fine-tune the dialogue act predictions, outperform past models for human-scored response quality and conversation engagement.
While this is a fairly straightforward idea with a long history, the authors claim to be the first to use dialogue act prediction for open-domain (rather than task-driven) dialogue. If that claim to originality is not contested, and the authors provide additional assurances to confirm the correctness of the implementations used for baseline models, this article fills an important gap in open-domain dialogue research and suggests a fruitful future for structured prediction in deep learning-based dialogue systems.

Some points:
1. The introduction uses "scalability" throughout to mean something closer to "ability to generalize." Consider revising the wording here.
2. The dialogue act tag set used in the paper is not original to Ivanovic (2005) but derives, with modifications, from the tag set constructed for the DAMSL project (Jurafsky et al., 1997; Stolcke et al., 2000). It's probably worth citing some of this early work that pioneered the use of dialogue acts in NLP, since they discuss motivations for building DA corpora.
3. In Section 2.1, the authors don't explicitly mention existing DA-annotated corpora or discuss specifically why they are not sufficient (is there e.g. a dataset that would be ideal for the purposes of this paper except that it isn't large enough?)
3. The authors appear to consider only one option (selecting the top predicted dialogue act, then conditioning the response generator on this DA) among many for inference-time search over the joint DA-response space. A more comprehensive search strategy (e.g. selecting the top K dialogue acts, then evaluating several responses for each DA) might lead to higher response diversity.
4. The description of the RL approach in Section 3.2 was fairly terse and included a number of ad-hoc choices. If these choices (like the dialogue termination conditions) are motivated by previous work, they should be cited. Examples (perhaps in the appendix) might also be helpful for the reader to understand that the chosen termination conditions or relevance metrics are reasonable.
5. The comparison against previous work is missing some assurances I'd like to see. While directly citing the codebases you used or built off of is fantastic, it's also important to give the reader confidence that the implementations you're comparing to are the same as those used in the original papers, such as by mentioning that you can replicate or confirm quantitative results from the papers you're comparing to. Without that there could always be the chance that something is missing from the implementation of e.g. RL-S2S that you're using for comparison.
6. Table 5 is not described in the main text, so it isn't clear what the different potential outputs of e.g. the RL-DAGM system result from (my guess: conditioning the response generation on the top 3 predicted dialogue acts?)
7. A simple way to improve the paper's clarity for readers would be to break up some of the very long paragraphs, especially in later sections. It's fine if that pushes the paper somewhat over the 8th page.
8. A consistent focus on human evaluation, as found in this paper, is probably the right approach for contemporary dialogue research.
9. The examples provided in the appendix are great. It would be helpful to have confirmation that they were selected randomly (rather than cherry-picked).

---

> ### Author Response · Authors · 2017-12-24
> **Rebuttal**
>
> Thank you for your valuable comments.
>
> 1.	We replace the word "scalability" in Introduction with other words (e.g., "scale to new domains").
>
> 2.	We follow your suggestions and cite related work about dialogue acts at the beginning of Section 2.1. We also mention a public DA corpora "the Switchboard Corpus" in the second paragraph of Section 2.1 and clarify that we build a new data set because no one has analyzed open domain dialogues with dialogue acts about conversational context before.
>
> 3.	We modify Equation (4) in the previous version as Equation (4)+Equation (5). Now dialogue act selection in our model becomes more general and takes multiple strategies (top 1 and top K) as special cases. We can try dialogue generation with top K acts in our future work.
>
> 4.	We break up some long paragraphs in Section 3.2 for ease of reading and cite (Li et al., 2016b) before the termination strategies, as some of them (e.g., regarding to repetitive turns) are inspired by the work.
>
> 5.	Although we use a different data set, the average number of turns of the simulated dialogues from RL-S2S in our work is very close to the number reported in (Li et al., 2016b). Our number is 4.36 (refer to the machine-machine column in Table 4(b)), while the number reported in (Li et al., 2016b) is 4.48. This might provide an additional evidence to the correctness of the implementation of the baseline model in the work.
>
> 6.	Table 5 in the previous version becomes Table 6 now. We describe the table right after it.  Basically, SL-DAGM and RL-DAGM share the same text generation but differs on how they select dialogue acts, as we only optimize the policy network with RL. The response given by RL-DAGM comes from CS.Q (clarified after the generated response in Table 6), while the response given by SL-DAGM comes from CS.S. Both are top dialogue acts under the corresponding policy networks.
>
> 7.	We follow your suggestions and break up long paragraphs.
>
> 8.	The examples given in Appendix are picked randomly.

---

### Official Review · AnonReviewer1 · 2017-11-28
**Interesting topic discussed, but the paper is not good enough**

**Rating:** 4
**Confidence:** 5

**Review:**

The topic discussed in this paper is interesting. Dialogue acts (DAs; or some other semantic relations between utterances) are informative to increase the diversity of response generation. It is interesting to see how DAs are used for conversational modeling, however this paper is difficult for me to follow. For example:

1) the caption of section 3.1 is about supervised learning, however the way of describing the model in this section sounds like reinforcement learning. Not sure whether it is necessary to formulate the problem with a RL framework, since the data have everything that the model needs as for a supervised learning.
2) the formulation in equation 4 seems to be problematic
3) "simplify pr(ri|si,ai) as pr(ri|ai,ui−1,ui−2) since decoding natural language responses from long conversation history is challenging" to my understanding, the only difference between the original and simplified model is the encoder part not the decoder part. Did I miss something?
4) about section 3.2, again I didn't get whether the model needs RL for training.
5) "We train m(·, ·) with the 30 million crawled data through negative sampling." not sure I understand the connection between training $m(\cdot, \cdot)$ and the entire model.
6) the experiments are not convincing. At least, it should show the generation texts were affected about DAs in a systemic way. Only a single example in table 5 is not enough.

---

> ### Author Response · Authors · 2017-12-24
> **Rebuttal**
>
> Thank you for your valuable comments.
>
> 1.    Why we need reinforcement learning
>
> As we have mentioned in the paper, open domain dialogue generation needs to be optimized for long-term engagement in practice.  Yes, in supervised learning, we have large scale of human dialogues tagged with dialogue acts, but that does not mean the algorithm can learn how to keep conversation going from the data, as more than 45% training dialogues are not longer than 5 turns (described in the last paragraph of Section 3.1 in the new version).  Supervised learning just learns a model by maximizing the likelihood of the observed data including the short dialogues. Dialogue acts are learned only according to the history, and no information of the future influence can flow in.  Then, without an additional objective (i.e., Equation (8) in Section 3.2) and mechanism (optimizing for future success), how can we (explicitly) guarantee that the model is optimized for long-term engagement? Therefore, supervised learning is to learn human language and reinforcement learning is to further optimize the combination of dialogue acts in order to achieve long-term conversation.
>
> Model optimization with reinforcement learning is also encouraged by the experimental results. In Table 5, response diversity is significantly improved by RL (see the difference between RL-DAGM and SL-DAGM on distinct-1 and distinct-2), and in Table 4(b), with RL, both the dialogues from machine-machine simulation and human-machine test become longer. Moreover, as we have analyzed in the last paragraph of Section 4.3, it is because RL can promote context switch in interactions that the model, after optimized with RL, can lead to better engagement. All the results well support our motivation to learning with RL.
>
> 2.     >>> the formulation in equation 4 seems to be problematic
>
> Thanks for pointing out this problem. We have modified Equation (4) in the previous version as Equation (4)+Equation (5) in the new version. Now the procedure of generation becomes more clear.
>
> 3.     >>>"Simplify pr(ri|si,ai) as pr(ri|ai,ui−1,ui−2) since decoding natural language responses from long conversation history is challenging" to my understanding, the only difference between the original and simplified model is the encoder part not the decoder part. Did I miss something
>
> Yes, from a model perspective, the simplification here just changes the encoder. However, what we mean here is that it is difficult for an RNN to memorize long conversation history, and thus encoding long history means either the response given by the decoder is irrelevant to the early history, or the response will be messed up.
>
> 4.     >>>"We train m(·, ·) with the 30 million crawled data through negative sampling." not sure I understand the connection between training $m(\cdot, \cdot)$ and the entire model
>
>  $m(\cdot, \cdot)$ is pre-trained and used to estimate the reward function in Equation (9). This is the only connection between  $m(\cdot, \cdot)$ and the entire model. We have clarified this in the paragraph after Equation (9).
>
> 5.    >>>the experiments are not convincing. At least, it should show the generation texts were affected about DAs in a systemic way. Only a single example in table 5 is not enough.
>
> Thanks for your comments. We do three things to show how the generated texts are affected by dialogue acts:
>
> (1)	We move Table 7 in the previous version from Appendix to Section 4.2. Now the table is Table 5. In the table, one can see that with dialogue acts, the diversity of generated responses is significantly improved (corresponding to much larger distinct-1 and distinct-2).  In the following explanation (the third paragraph of Section 4.2), we claim that this is one benefit of dialogue acts, as search space now becomes act × language.
>
> (2)	We add Section 4.4 where we compare responses from different dialogue acts using some metrics. The conclusion is that responses generated from CS.* are longer, more informative, and contain more new words than responses generated from CM.*, and statements and answers are generally more informative than questions in both CS.* and CM.*.  Please refer to the new version of the paper to get more details.
>
> (3)	In the last paragraph of Section 4.3, we show that simulated dialogues without CS.* are much shorter than those with CS.* (SL: 4.78 v.s. 8.66, RL: 2.67 v.s., 8.18). The result indicates that if we remove CS.*, then the conversation engagement of our model may degrade to the baseline model.

---

### Official Review · AnonReviewer4 · 2017-11-29

**Rating:** 7
**Confidence:** 4

**Review:**

The paper describes a technique to incorporate dialog acts into neural conversational agents.  This is very interesting work.  Existing techniques for neural conversational agents essentially mimic the data in large corpora of message-response pairs and therefore do not use any notion of dialog act.  A very important type of dialog act is "switching topic", often done to ensure that the conversation will continue.  The paper describes a classifier that predicts the dialog act of the next utterance.  The next utterance is then generated based on this dialog act.  The paper also describes how to increase the relevance of responses and the length of conversations by self reinforcement learning.  This is also very interesting.  The empirical evaluation demonstrates the effectiveness of the approach.  The paper is also well written.  I do not have any suggestion for improvement.  This is good work that should be published.

---

> ### Author Response · Authors · 2017-12-24
> **Rebuttal**
>
> Thank you for your comments

---

### Author Response · Authors · 2017-12-24
**A new version of the paper has been uploaded**

We upload a new version of the paper in which we try our best to address the concerns from the reviewers.
Major revisions include:
(1) We cited more work about dialogue acts in Section 2.1, and commented on the existing dialogue act corpus.
(2) We changed Equation (4) in the previous version to Equation (4) and Equation (5). Now, the mechanism of dialogue generation becomes more clear and general.
(3) We further justified the necessity of reinforcement learning at the end of Section 3.1 with the distribution of the data set.
(4) To emphasize the effect of dialogue acts to generation, we moved Table 7 in  Appendix in the previous version to the main text. Now the table becomes Table 5.
(5) We added Section 4.4 where we systematically analyze how the generated text is affected by the dialogue acts with some automatic metrics.
(6) We broke up some long paragraphs for ease of reading.

---

### Public Comment · (anonymous) · 2018-01-14
**Similar idea from ACL'17**

The idea in this paper looks very similar to the idea from <Learning Discourse-level Diversity for Neural Dialog Models using Conditional Variational Autoencoders> which was presented at ACL'17: https://arxiv.org/abs/1703.10960. Especially, the idea of using dialog act in open domain dialogue generation, which is the main contribution of this paper, was firstly introduced in https://arxiv.org/abs/1703.10960

I'd like the authors to clarify how they differ and would like to ask the reviewers to read https://arxiv.org/abs/1703.10960 and see how this affects your judgment of the submission.

---

> ### Author Response · Authors · 2018-01-14
> **Difference lies in four aspects: goal,  dialogue acts,  model, and learning method**
>
> Thank you for leading us to the ACL paper.
>
> While overall the ACL paper is about how to model dialogues with latent variables using VAE techniques, which is very similar to the early work of Serban et al., https://arxiv.org/abs/1605.06069 (VHRED, this is also a baseline in our experiment), the paper does cover dialogue acts from an existing data set, the Switchboard Corpus, in the experiment part as an extra feature.
>
> We would say that the ACL paper is about how to generate a reply for a static context. It is NOT a study of how dialogue acts can be used to interpret and control dynamic human-machine interactions as policies, NOR it is a study of how the generated responses will be affected by the dialogue acts, which are the major contributions of our work.
>
> Here, we would like to clarify the  originality (contributions) of our paper and the difference with the ACL paper.
>
> ORIGINALITY:
> 1. The first study of how to design dialogue acts to understand how human behave and engage in social chat, instead of using dialogue acts to ground semantics of utterances. The contribution lies in filling the gap between end-to-end open domain dialogue modeling and task-oriented dialogue modeling with task specific dialogue acts, which is not covered by the ACL paper.
>
> 2. Insights from the analysis of human interactions. The contribution lies in discovery of the role of context switch for engagement in social chat with quantitative evidences, which is not covered by the ACL paper.
>
> 3. The first work on modeling the policy in open domain dialogue management with dialogue acts  and learning the policy (i.e., the combination of dialogue acts in a conversation flow) for long-term conversation by reinforcement learning, which is not covered by the ACL paper.
>
> 4. Empirical studies of how human-machine conversation is affected by the dialogue acts, from static generated text to dynamic interactions, which is not covered by the ACL paper.
>
>
> DIFFERENCE
> 1. Goal.  The goal of the ACL paper is to study how to leverage VAE techniques to address the"one-to-many" problem in open domain dialogues. The major contribution of the ACL paper also lies in the VAE framework, as summarized by the authors in the last paragraph of Introduction of the paper. Our goal is to INTERPRET engagement in human-human social interactions with dialogue acts and thus ENHANCE human-machine engagement in open domain conversations by combining the dialogue acts.
>
> 2. Dialogue acts.  Dialogue acts  in ACL paper  are extra features, therefore they come from an existing data set and follow a traditional scheme (42 acts).  Our dialogue acts, however, are designed to describe how human perform in order to keep their social interactions. Therefore, they highly connect to human behavior regarding to conversational contexts and are not covered by any of the existing data sets.  Dialogue acts in the ACL paper are used to measure how well the generated responses are and justify  that the learned latent representations are reasonable. Our dialogue acts, however, are used to interpret and control the flow of human-machine interactions in order to achieve long-term conversation.
>
> 3. Model. The ACL paper deals with static response generation (i.e., given a fixed context, how to generate a proper response). Therefore, only the  dialogue act of the last turn (in training) and the predicted dialogue act of the response (in both training and test) are treated as extra features in latent representation learning and response decoding. There are no studies on how the dialogue acts are coordinated and thus affect the flow of conversation. While we deal with dynamic human-machine interactions, therefore, dialogue acts are used to model the policy of dialogue management and guide the flow of conversation. Coordination of dialogue acts across multiple turns is implicitly modeled in the policy network. Note that we also show results of static response generation. That is just to show that our model can give reasonable intermediate results in interactions.
>
> 4.  Learning method. The ACL paper learns a dialogue model with VAE. While we treat dialogue acts as a kind of strategies and use reinforcement learning to enhance engagement in human-machine interactions.
>
> Although there exists significant difference, we would like to show our respect to this ACL work.  We upload a new version and make the following changes:
> 1. We cite the paper and clarify the difference in Related Work
> 2. We position our contributions on dialogue act design for interpretation and learning dialogue policies with the dialogue acts.
> 3. We withdraw the claim of “first work of using dialogue act in open domain dialogue generation”, and change it  to “we are the first who design dialogue acts to explain social interactions, control open domain response generation, and guide human-machine conversations.” (see the first paragraph of Related Work)

---

### Author Response · Authors · 2018-01-19
**New updates**

We updated the paper according to the comments about the existing work.

Specifically, in the new version, we made the following updates:

1. We withdraw the claim of "first work of using dialogue act in open domain dialogue generation" and position our work as “we are the first who design dialogue acts to explain social interactions, control open domain response generation, and guide human-machine conversations.” (see the first paragraph of Related Work)

2. We cite the ACL paper and clarify the difference from it at the end of the first paragraph of Related Work.

3. We further emphasize our motivation on using RL in the last paragraph of Section 3.1 and the first paragraph of Section 3.2

4. We correct some typos. For example: HERD-> HRED, VHERD->VHRED.

---

### Public Comment · (anonymous) · 2018-01-22
**missing reference?**

Besides the ACL 2017 paper that has been mentioned before, another recent paper that seems relevant is Serban et al 2017, A Deep Reinforcement Learning Chatbot, https://arxiv.org/abs/1709.02349, which also uses dialogue acts to learn a policy in a bot-human conversation setting. Can the authors comment on how their submission relates to this prior work?

---

> ### Author Response · Authors · 2018-01-23
> **Comments on difference (we cited the work in the new version)**
>
> Thank for reminding us the paper.
>
> This is a system description paper in which the system is the one built by MILA for the 1st Alexa Prize competition. In the system,  dialogue acts are used as one of the 1458 features  for learning a response selection model and a special feature in the following abstract discourse MDP model.
>
> Because the work is building a system for competition, it is about ensemble of response providers from multiple sources. Therefore,  the policy is response select strategy from a bunch of candidates given the existing contexts. For details, please refer to the second paragraph of Section 4 of the paper.    We would say dialogue acts are not used to guide the generation of responses as most responses in the related work are provided by other sources and it only cares how to select one from the candidates, nor the work used dialogue acts as policies to manage the flow of conversation. Dialogue acts in the work are extra features in the learning of response selection, even in the abstract discourse MDP.  As an evidence, both the states and the actions in their models (Equation (5) and Equation (20)) are not related to dialogue acts, but are conversation history and responses respectively. In abstract discourse MDP, dialogue acts are used as one of the 3 features to guide training data sampling (Equation (18)) for the learning of the response selection model.  Although the work aims to achieve engagement in social chat, it provides no insights on how to understand engagement with dialogue acts, not to mention imitating such behaviors with dialogue acts.
>
> In summary, we would say the same thing as we commented on the ACL work:
> It is NOT a study of how dialogue acts can be used to interpret and control dynamic human-machine interactions as policies, NOR it is a study of how the generated responses will be affected by the dialogue acts, which are the major contributions of our work.
>
> Details:
> 1.  Dialogue acts. The dialogue acts in the related work mainly come from the existing 42 acts in Switchboard Corpus, which reflect semantics of single utterances.   Our dialogue acts, however, are designed to describe how human perform in order to keep their social interactions. Therefore, they highly connect to human behavior regarding to conversational contexts. The dialogue acts in the related work are used as features for response selection, while our dialogue acts are used to explain human interactions and control the generation of responses in order to keep conversation going.
>
> 2. Insights. No insights on what leads to engagement are provided in the related work. The work just reported  evaluation results in the competition.  We show enough insights from the analysis of human interactions and find that  context switch and question are key acts for engagement in social chat with quantitative evidences.
>
> 3. Model. The related work learns a response selection model, while we learn response generation. Therefore, we can show how different types of responses can be generated regarding to different dialogue acts.
>
> 4.  Learning. The related work optimizes according to human ratings (from both AMT and Alexa users), while we optimize our model by encouraging long and reasonable conversations through self-play, which is more cost-effective.
>
> Again, we would like to show our respect to the work and have cited it in the new version

---

### Decision · Program_Chairs · 2018-01-29
**ICLR 2018 Conference Acceptance Decision**

**Decision:**

Reject

**Comment:**

This work takes dialogue acts into account to generate responses in a human-machine conversation. However, incorporating dialogue acts into open-domain dialogue was already the focus of Zhao et al's ACL 2017 paper, Learning Discourse-level Diversity for Neural Dialog Models using Conditional Variational Autoencoders, and using dialogue acts in a policy for human-machine conversation was also an idea that already appeared in Serban et al 2017, A Deep Reinforcement Learning Chatbot. Despite the authors' response that tries to adjust their claims and incorporate a more thorough overview, I encourage the authors to re-work their research with a much more careful and reliable examination of previous work and how their effort should be understood in that more comprehensive context.